# Incidence and predictors of HIV related opportunistic infections after initiation of highly active antiretroviral therapy at Ayder Referral Hospital, Mekelle, Ethiopia: A retrospective single centered cohort study

Zekarias Gessesse Arefaine[1]*, Sintayehu Abebe[1], Ephrem Bekele[1], Amir Adem[1], Yohannes Adama[2], Norbert H. Brockmeyer[3], Judith Coenenberg[3], Anja Potthoff[3], Teferi Gebru Gebremeskel[4]

1 Department of Internal Medicine, College of Health Science, Mekelle University, Mekelle, Ethiopia,
2 Department of Public Health, College of Health Science, Mekelle University, Mekelle, Ethiopia, 3 Clinic for Dermatology, Venerology and Allergology of the Ruhr-Universität Bochum, Bochum, Germany,
4 Department of Reproductive Health, College of Health Sciences, Aksum University, Aksum, Ethiopia

* gessesse359@yahoo.com

## Abstract

### Introduction

Even though use of antiretroviral therapy (HAART) decreases the incidence of opportunistic infections (OIs) they are continuing to be a major cause of morbidity and mortality. Studies concerning this problem are scarce in Eastern Africa. The aim of this study was to determine the incidence and predictors of OIs after initiation of HAART in Ethiopia.

### Methods

A health facility based single centered cohort study using structured data extraction sheet was conducted. The study population was all HIV positive ART naive adolescents and adults who started HAART between January 2009 and May 2012. Simple random sampling technique was used to select 317 patients from the record. Multivariate binary logistic regression model was used to determine factors for the occurrence of OIs after initiation of HAART.

### Results

The incidence of OIs after HAART was 7.5 cases/100person years. Tuberculosis, oral candidiasis, pneumonia and toxoplasmosis were the leading OIs after HAART. A bed ridden functional status at initiation of HAART, presence of OIs before HAART, non-adherence and low hemoglobin level were predictors for the occurrence of OIs after HAART.

**Data Availability Statement:** All relevant data are within the paper and its Supporting Information files.

**Funding:** The authors received no specific funding for this work.

**Competing interests:** The authors declare that they have no competing interests.

## Conclusion

The incidence of OIs after HAART was higher than in previous studies. Patients with the identified risk factors need strict follow up to reduce the morbidity and mortality attributed to OIs. Earlier initiation of HAART before advanced immune suppression, better management of TB and extended baseline assessment could help to reduce opportunistic infections and mortality after the initiation of HAART in Ethiopian patients.

## Introduction

Before the widespread use of High Active Antiretroviral Therapy (HAART), opportunistic infections (OIs), which have been defined as infections that are more frequent or more severe due to immune suppression were the principal cause of morbidity and mortality in HIV patients [1]. OIs that occur after initiation of HAART are categorized into three groups; the first group includes OIs that occur shortly after the start of HAART (within 12 weeks) because of worsening of previously diagnosed OIs or unmasking of subclinical infection. These cases represent immune reconstitution inflammatory syndrome (IRIS). The second group are OIs that occur >3 months after initiation of HAART in patients with suppressed viral load and sustained CD4 count >200cells/mm$^3$. Determining whether it is IRIS or incomplete immunity with the occurrence of new OIs is difficult. The third group includes OIs that occur in patients with virology and immunologic failure on HAART [1, 2, 3, 4]. According to the country progress report on HIV/AIDS of 2012, in Ethiopia at the end of 2011 the total number of adults in need of HAART was 333,434 out of which 249,174 (74.7% of eligible according to WHO criteria) were started on HAART [5]. In Ethiopia, a study done by Huruy K. et.al in Zewditu Memorial Hospital showed that the proportion of OIs as immune restoration disease after initiation of HAART is 10.6% [6]. Several studies were conducted in different parts of the world concerning occurrence of OIs following initiation of HAART [4]. In our setting studies related to this problem are limited. The aim of this study is to determine incidence and risk factors for occurrence of OIs following HAART initiation.

## Methodology

The study was conducted in Ayder Referral Hospital, teaching hospital for the College of Health Sciences, Mekelle University, which is located in the city of Mekelle, Tigray Region (North Ethiopia). The Hospital commenced rendering its referral and specialized medical services in 2008 to the 8-million populations in its catchment areas of the Tigray, Afar and Southeastern parts of the Amhara Regional States. It provides a broad range of medical services to both in and out patients. As such, the hospital can be designated as the most advanced medical facility, by all accounts, in the northern part of the country and that it stands as the second largest hospital in the nation [7].

Health facility based retrospective single-center cohort study was conducted among HIV positive patients to assess the incidence and risk factors for HIV related OIs after initiation of HAART. The study period was from January, 2009 to May 2012. Data collection was carried out from July 15th to July 30th 2012. All HIV positive ART naïve adolescents and adults ever started HAART in Ayder Referral Hospital were included. Initiation of ART in the HIV-positive patients followed WHO recommendations: all patients with WHO stage IV disease, WHO stage III disease with a CD4 cell count <350 cells/μl or with a CD4 cell count <200 cells/μl

were eligible for ART. An ethical clearance letter was obtained from Mekelle University College of Health Science (CHS) ethical review committee and permission was obtained from CHS dean office before data collection. Any identifier information was excluded and confidentiality was kept during data collection. Sample size was determined using a formula for comparison of proportions. Internal comparison was done using a low baseline CD4 count as a predictor for occurrence of OIs after initiation of HAART. A total of 317 HIV positive adolescents and adults were selected using simple random sampling from 649 patients who started HAART during the period of January 2009 to May 2012. The samples were taken randomly using Excel Rand function. Data was collected from ART registry log-book and patients chart. Data extraction sheet was prepared and used to collect the data. The data was collected by health professionals who have training on HAART. After being coded it was entered to a data sheet of SPSS version 16.0. Multivariate binary logistic regression model was used to determine factors for the occurrence of OI after initiation of HAART. Kaplan-Meier was used to estimate OI free survival time after HAART initiation. Odds ratio, and P- value were determined to check association between variable and p-value <0.05 considered significant.

Ethical clearance was obtained from the Institutional Review Committee(IRC) of College of Medicine and Health Sciences mekelle university. Permission letter was received from those administrative bodies of Ayder comprehensive specialized hospital, then proceeded with data collection.

## Results

A total of 317 charts of patients ever having started HAART were included in the retrospective analysis. 184 (58%) patients were female; male to female ratio was 1:1.38. The mean age was

**Table 1. Socio-demographic characteristic of patients started HAART at Ayder Referral Hospital, Mekelle, Ethiopia.** (n = 317) January 2009 to May 2012.

| Characteristics | Frequency | Percent |
|---|---|---|
| Age group | | |
| 16–28 | 5 | 1.6 |
| 29–34 | 24 | 7.6 |
| 35–40 | 235 | 74.1 |
| 41–54 | 39 | 12.3 |
| 55–78 | 14 | 4.4 |
| Sex | | |
| Male | 133 | 42 |
| Female | 184 | 58 |
| Marital status | | |
| Single | 73 | 24.2 |
| Married | 150 | 49.7 |
| Divorced | 46 | 15.2 |
| Widowed | 13 | 4.3 |
| Separated | 20 | 6.6 |
| Employment Status | | |
| Unemployed | 193 | 60.9 |
| Working | 124 | 39.1 |
| Educational status | | |
| No education | 97 | 31.8 |
| Primary | 114 | 37.4 |
| Secondary | 65 | 21.3 |
| Tertiary | 29 | 9.5 |

34.8±8.9 years. The minimum and maximum age was 16 and 78 years respectively. The majority of patients (74.1%) were in the age group 35–40 years (Table 1).

Half of the patients were married, 15.5% divorced, 4.3% widowed and 24.2% single. The majority of the patients (60.9%) were unemployed and 39.1% were working at initiation of HAART.

As to the functional status of patients recording was traced for 314 patients, the majority was fit for work (60.8%), ambulatory was 27.7% and bed ridden 11.5%. The mean body weight at initiation of HAART was 48.9±9.5 kg, which is less than the average weight of an Ethiopian (63.5–72.5 kg) [8]. At initiation of HAART the mean hemoglobin level was 11.9 g/dl with minimum hemoglobin of 4.5g/dl and maximum of 19.1g/dl, 35.6% of the patients were anemic at initiation of HAART using WHO definition of anemia. Half of the patients (52.4%) were in WHO clinical stage III and 23.7% in stage IV. Before initiation of HAART 43.2% of the patients had opportunistic infections. Among the spectrum of OIs occurring before HAART, tuberculosis accounts for half of the opportunistic illnesses with 50.4% followed by oral candidiasis 15.6% (Table 2).

The majority of patients (86.5%) who had OIs before HAART had a $CD4^+$ count of <200 cells/$mm^3$. At the time of HIV diagnosis 80.1% of patients had a $CD4^+$ cell count of <200 cells/$mm^3$. At initiation of HAART 22.1% of the patients had a $CD4^+$ count of <50 cells/$mm^3$, 61.8% of the patients had $CD4^+$ count of 50–199 cells/$mm^3$. A total of 83.9% had a $CD4^+$ count of <200cells/$mm^3$, 14.2% of 200–349 cells/$mm^3$ and 1.9% of >350 cells/$mm^3$. The mean $CD4^+$ cell count at initiation of HAART was 121±81 cells/$mm^3$ (5 to 442 cells/$mm^3$). The proportion of patients who had $CD4^+$ cell count <200 cells/$mm^3$ decreased from 83.9% at initiation of HAART to 23% at 6 months, 7.9% at 12 months, and 1.6% at 18 months after HAART initiation.

Almost all (99.7%) patients started with Co-trimoxazole preventive therapy. INH (Isoniazid preventive therapy) was started for 21.1% of patients. INH preventive therapy was not prescribed for 26.5% of the patients due to past or current tuberculosis, but for 51.4% of the patients the reason for not prescribing INH was not documented. Initial HAART regimens used were AZT+3TC+NVP (29.7%), TDF+3TC+EFV (29%), D4T+3TC+NVP (20.2%), TDF +3TC+NVP (8.8%), D4T+3TC+EFV (7.6%) and AZT+3TC+EFV (4.7%). Adverse drug reactions were reported or documented in only 7.2% of the patients. The side effects reported were rash (21.7%), anemia (21.7%), nausea (13%), numbness (4.3%) and jaundice (4.3%). During

**Table 2. Spectrum of HIV related OIs before HAART, at Ayder Referral Hospital, Mekelle, Ethiopia.** (n = 317) January 2009 to May 2012.

| List of OIs before HAART | Frequency | Percent |
|---|---|---|
| Tuberculosis | 69 | 50,4 |
| (Disseminated+ pulmonary+ Lymphadenitis+ meningitis) | (33+ 27+ 7 +2) | |
| Oral candidiasis | 22 | 15.6 |
| Pneumonia | 13 | 9.6 |
| CNS Toxoplasmosis | 12 | 8.9 |
| Chronic diarrhea | 7 | 5.2 |
| PCP | 7 | 5.2 |
| Herpes zoster | 4 | 3.0 |
| Herpes genitalia | 1 | 0.7 |
| Visceral Leishmaniasis | 1 | 0.7 |
| Esophageal candidiasis | 1 | 0.7 |
| **Total** | **137** | **100** |

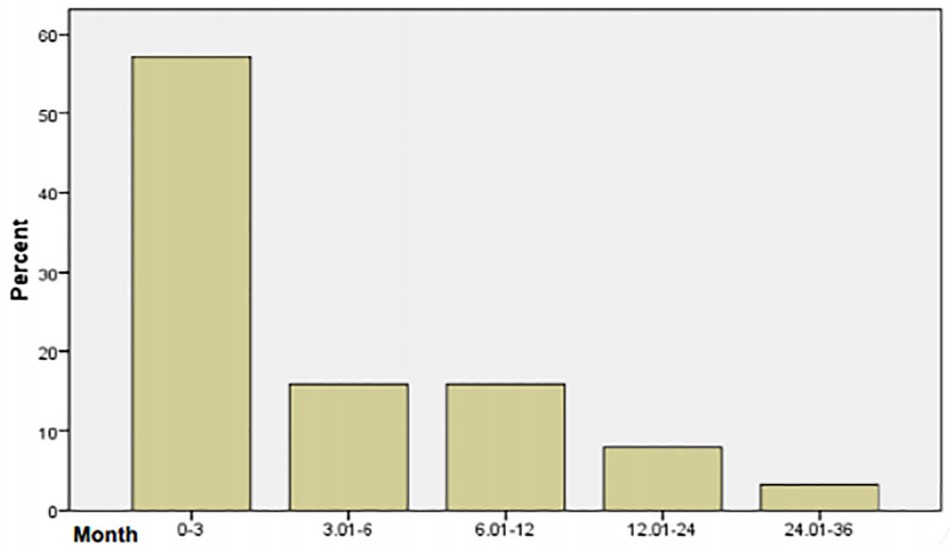

**Fig 1. Months AIDS related OIs occurred after initiation of HAART.**

treatment 91.8% had good adherence, 5.6% had fair adherence, 2.6% had poor adherence using patient´s self-report as method of adherence assessment and definition of adherence (good adherence $\geq$ 95%, fair adherence 85% to 95% and poor adherence $\leq$ 85%).

The incidence of OIs after two weeks of initiation of HAART was 64 (20.2%) in 855 person years of follow-up or incidence density of 7.5 cases/100 person years. The duration of follow up time after HAART ranged from two weeks to 36 months. The number of OIs per patient ranged from none to three. At 3 months after HAART initiation the incidence of HIV related OIs was 48.2 per 100 person years, at 6 months 31.2 per 100 person years, at 12 months 19.3 per 100 person years, at 24 months 10.6 per 100 person years and at 36 months 7.5 per 100 person years. The majority of patients (57.1%) developed OIs within the first 3 months after HAART (Fig 1).

The common OIs seen after initiation of HAART were tuberculosis (32.3%), oropharyngeal candidiasis (15.3%), pneumonia including pneumocystis jiroveci pneumonia (13,8) and CNS toxoplasmosis (13,8%) (Table 3).

The diagnosis of OIs were made clinically in 45.3% of the patients, both clinical and laboratory in 43.8% and laboratory methods alone were used in 10.9% of the patients with OIs. Methods utilized were x-ray (52.9%), fine needle aspiration (20.3%), body fluid analysis (14.7%), x-ray and ultrasound (5.9%), sputum examination (2.9%) and endoscopy (2.9%).

At six months following HAART initiation the mean CD4$^+$ cell count was 250±111/µl for 46.5% of the patients which is a statistically significant elevation from the value at initiation of HAART (P = 0.007). After occurrence of OIs 60.3% of the patients improved and are still on treatment, 28.6% died, 6.3% were transferred to other health facilities for follow up and the status of 4.8% of the patients was unknown.

In Kaplan-Meier, the median OI free survival time after initiation of HAART was two months with 95% confidence interval of 1.2–2.9 months. When the analysis was stratified with gender, the median OI free survival time after initiation of HAART for males and females was 3.3 and 1.8 months respectively, but the difference was not statistically significant.

In bivariate analysis a bed ridden functional status at initiation of HAART (OR: 3.8, 95% CI 1.7–8.4, P: 0.001), and WHO clinical stage IV disease (OR: 6.5, 95%, CI: 1.4–29.7, P: 0.01) were predictors for occurrence of OIs after HAART.

**Table 3. List of HIV related OIs diagnosed after two weeks of HAART initiation at Ayder Referral Hospital, Mekelle, Ethiopia.** (n = 317) January 2009 to May 2012.

| List of OIs after initiation of HAART | Frequency | percent |
|---|---|---|
| Tuberculosis | 21 | 32.3 |
| (Dissiminated, pulmonary, lymphadenitis, meningitis) | (10+ 5+ 4+ 2) | |
| Oral candidiasis | 10 | 15.3 |
| Pneumonia | 9 | 13.8 |
| CNS toxoplasmosis | 9 | 13.8 |
| PCP | 4 | 6.2 |
| Herpes Zoster | 3 | 4.6 |
| Visceral leishmaniasis | 2 | 3.1 |
| Cryptococcus meningitis | 2 | 3.1 |
| Pyogenic meningitis | 2 | 3.1 |
| Esophageal candidiasis | 1 | 1.5 |
| Chronic diarrhea | 1 | 1.5 |
| Sepsis | 1 | 1.5 |
| **Total** | **65** | **100.0** |

In multivariate analysis presence of OIs before initiation of HAART (OR: 2.8, 95% CI1, 6.9, P: 0.021), non-adherence (fair, poor) for HAART during their follow up period (OR: 14.6, CI: 5.8–119, P: 0.000) and hemoglobin level of less than 7gm/dl at initiation of HAART (OR: 6.8, P: 0.02) were independent predictors for occurrence of OIs after HAART. A CD4$^+$ cell count of less than 200 cell/mm$^3$ at initiation of HAART was associated with occurrence of OIs after HAART but the association was not statistically significant (OR: 1.6, P: 0.34) (Table 4).

## Discussion

This study aimed to determine incidence and risk factors for OIs following the initiation of HAART in Ethiopia. Patients´ socio-demographic characteristics showed the typical distribution of an African setting similar as described by the Antiretroviral Therapy in Lower Income Countries (ART-LINC) collaboration [9]. The majority of the patients were women and between 30 and 40 years old.

The incidence of OIs after initiation of HAART was 7.5 cases/100 person years and half of it occurred within the first 3 months. This finding is higher than the incidence of HIV related OIs after initiation of HAART seen in different studies conducted in Zewditu Memorial Hospital Ethiopia (14.6%), the United States (16.6%), France (11.3%), and India (8.3%) [10, 11, 12, 13]. The study methods used in our study were similar to those of the study from Zewditu Memorial Hospital except the fact that the study from Zewditu Memorial Hospital considered only immune restoration diseases, which are defined as occurrence of HIV related OIs shortly after the start of HAART (within 12weeks). In the above mentioned prospective studies conducted in the United States, France and India a prospective cohort was established. Baseline characteristics were different in the US and France with higher CD4 cell counts (266/µl and 298 respectively) at initiation of HAART and more men who had sex with men included. The Indian study included a profound baseline screening. The incidence of OIs after HAART in our study was also higher in comparison to the study conducted in Poland (between 2000 and 2002), where the incidence of AIDS defining illness was 6.8 in 2000, 6.5 in 2001 and 4.8 100 persons/year in 2002. The incidence of OIs after HAART in our study was lower than compared to the prospective cohort study conducted in Senegal, which showed an incidence of 30% [14, 15].

**Table 4. Incidence of HIV related OIs after HAART in relation to potential predictor variables, at Ayder Referral Hospital, Mekelle, Ethiopia.** (n = 317) January 2009 to May 2012.

| Characteristics | AIDS related OIs after HAART | | | |
|---|---|---|---|---|
| **Sex** | No OIs (%) | OIs present | COR(95%CI) | AOR(95%CI) |
| Male | 101(31.7) | 32(10.1) | 1 | 1 |
| Female | 152(48.3) | 32(10.1) | 1.5(0.9–2.6) | 1.7(0.8–4) |
| **Educational status** | | | | |
| No education | 75(23.6) | 22(8.2) | 1 | 1 |
| Primary | 89(29.6) | 25(7.8) | 0.9(0.5–1.8) | 1(0.4–2.9) |
| Secondary | 57(18.5) | 8(2.5) | 0.5(0.2–1.2) | 0.4(0.1–1.5) |
| Tertiary | 21(6.9) | 8(2.5) | 1.3(0.5–3.3) | 3.4(0.7–16.2) |
| **Functional status** | | | | |
| Working | 164(51.7) | 27(9.1) | 1 | 1 |
| Ambulatory | 64(20.3) | 23(7.4) | 2.1(1.2–4)* | 2.1(0.5–9) |
| Bed ridden | 22(7) | 14(4.5) | 3.8(1.7–8.4)** | 3.9(1–13.9)* |
| **Hemoglobin level** | | | | |
| Severe anemia | 8(2.5) | 4(1.3) | 2.5(0.7–2.8.7) | 6.8(2–22.4)** |
| Moderate anemia | 12(2.2) | 3(2.5) | 1.2(0.3–4.6) | 0.9(0.2–6.2) |
| Mild anemia | 63(19.8) | 23(7.3) | 1.8(0.9–3.3) | 2.4(0.9–5.9) |
| Normal | 170(53.6) | 34(10.8) | 1 | 1 |
| **Base line CD4+** | | | | |
| <200 | 208(65.8) | 58(18.2) | 1.7(0.9–3.2) | 1.6(0.6–4.4) |
| 200–349 | 40(12.7) | 5(1.5) | 0.5(0.2–1) | 0.7(0.3–2.7) |
| >350 | 5(1.5) | 1(0.1) | 1 | 1 |
| **WHO clinical staging** | | | | |
| Stage 4 | 29(9.2) | 2(0.6) | 6.4(1–29.7)* | 1.2(0.2–9) |
| Stage 3 | 38(12.4) | 7(2.2) | 3.5(0.7–15.3) | 0.6(0.1–4.5) |
| Stage 2 | 134(42.2) | 32(10) | 2.6(0.5–13.4) | 0.5(0.07–4) |
| Stage 1 | 29 | 2 | 1 | 1 |
| **Adherence** | | | | |
| Good | 234(73.8) | 46(18) | 1 | 1 |
| Non-Adherent(fair, poor) | 10(3.5) | 15(4.7) | 7.6(3.2–18)** | 14.6(5.8–119)** |

*P value <0.05

** P value <0.001, COR- crude Odds ratio, AOR- Adjusted odds ratio

There are a number of predictive variables for the occurrence of OIs in our setting. Study patients who were bed ridden at the start of HAART were at increased risk of HIV related OIs after initiation of HAART (OR: 2.8, P: 0.02) in bivariate analysis.

Our study also revealed that presence of OIs at initiation of HAART was associated with an increased risk of OIs after HAART (OR: 2.8, P: 0.02) which is in agreement with the studies conducted in Europe, Senegal and Nigeria [12, 15, 16, 17].

Non adherence for HAART was the predictor for occurrence of OIs after HAART (OR: 14.6, P: 0.003) which is consistent with other studies in Europe and North America [17, 18, 19, 20, 21].

Patients with low hemoglobin level had a higher risk of developing HIV related OIs after HAART (OR: 11.6, P: 0.016) which is consistent with other studies [11, 14].

Most patients presenting in Ayder hospital were medically eligible for HAART according to WHO, at the time of HIV diagnosis, however for some patients WHO criteria did not allow

earlier start of treatment at the time of the study. Earlier treatment is now also recommended by the WHO [22]. In our study the majority of the patients (76.1%) were in advanced stage of HIV/AIDS (WHO clinical stage III and IV) at initiation of HAART. Before initiation of HAART 43.2% of the patients were diagnosed with HIV related OIs which was lower than in a study from India (68.5%) and Senegal (55%) [13, 15]. Insufficient diagnosis of OIs at baseline is a possible limitation of these data. Among the OIs occurring before HAART tuberculosis accounts for around half of OIs followed by oropharyngeal candidiasis, which is consistent with the study from India which showed tuberculosis as the leading OI followed by oral candidiasis [13].

In our study 83.9% of the patients had a $CD4^+$ count of $<200$ cells/mm$^3$ at initiation of HAART, which shows that patients had advanced immune suppression. The majority of patients (86.5%) who had OIs before HAART had a $CD4^+$ count of $<200$ cells/mm$^3$ which is slightly higher than in the Indian study by Srirangaraj et al. (79.5%) [13]. Surprisingly a low $CD4^+$ cell count at the initiation of HAART was not an independent predictor for OIs in our setting. In this study low baseline $CD4^+$ cell count did not have a statistically significant association with the occurrence of OIs after HAART (OR: 1.6, P: 0.34) but studies conducted in US by Buchacz et al. and the Swiss HIV Cohort Study revealed that low baseline $CD4^+$ cell count was a predictor for occurrence of OIs after HAART [11, 23]. This is probably due to the high number of tuberculosis infections and to the fact that only WHO stage III and IV patients with more than 200/µl CD4 could start HAART.

The leading HIV related OIs occurring after HAART in our study were tuberculosis followed by oral candidiasis which is in agreement with the study conducted in Zewditu Memorial Hospital which revealed tuberculosis as the leading OI (66.5%). The main HIV related OIs in the Senegalese study were recurrent pneumonia (37.3%), tuberculosis (26.2%), recurrent herpetic ulceration (21.7%), visceral candidiasis (15%), which still showed tuberculosis as one of the most common HIV related OIs after HAART. The incidence of OIs after HAART in the Indian study was 8.3% and the spectrum of OIs ranged from herpes zoster, non-typhoidal salmonella, cerebral toxoplasmosis, oral candidiasis to cryptosporidiosis, which is different from our study and from European settings [12, 23]. The consequent baseline screening for TB might explain the difference.

The rate of reduction in incidence of OIs in our study was 35% within 6 months, and 39% within 12 months. Compared to the Swiss cohort the rate of reduction in incidence was low in the first 6 months and higher after 12 months (66% at 6 months and 16% at 12 months) [23].

In our study 28% of the patients who started HAART died. 76% of the deaths occurred in the first 6 months after HAART initiation and tuberculosis is the leading cause of death, accounting for 64.6%. 84% of TB related deaths occurred in the first 6 months. In an Ethiopian study by Kassa et al. 57% of TB related deaths in HIV patients occurred within the first six months of HAART initiation [10]. Management of TB in patients starting HAART remains a challenge in our setting in Ayder Hospital, Ethiopia. IRIS could contribute to this and should be taken into account. New guidelines recommend delayed HAART start in patients with above 100/µl CD4. In the ART-LINC study one year mortality was 6,4% in low-income countries and mortality rate declined within the first few months of HAART [9].

According to a study conducted in France, old age ($>55$years) was a predictor for occurrence of OIs after HAART but in our study age was not a predictor for occurrence of OIs after initiation of HAART [11]. This could be due to the limited number of older patients ($>55$years) in our study, also reflecting the age distribution in the Ethiopian general population.

## Conclusion

HIV-patients with the risk factors bed ridden at initiation of HAART, presence of OIs before HAART, non-adherence for HAART, and low baseline hemoglobin level need strict follow up to reduce the morbidity and mortality attributed to OIs. In this study low baseline CD4$^+$ cell count did not have a statistically significant association with the occurrence of OIs, but high numbers of tuberculosis infections, lack of systematic initial OI screening and the fact that only WHO stage III and IV patients with less than 200/μl CD4$^+$ cell could start HAART might have influenced these data. Earlier initiation of HAART before advanced immunosuppression, better management of TB and extended baseline assessment could help to reduce opportunistic infections and mortality after the initiation of HAART in Ethiopian patients.

## Supporting information

**S1 File. Questionnaire English version.**
(DOCX)

**S2 File. Data set.**
(SAV)

## Acknowledgments

We are highly indebted to acknowledge Mekelle university college of health science, Rhur university, Bochum all and all participants of the study including supervisors of data collection and data collectors for their worthy efforts and participation in this study.

## Author Contributions

**Conceptualization:** Zekarias Gessesse Arefaine, Sintayehu Abebe, Ephrem Bekele, Amir Adem, Yohannes Adama, Norbert H. Brockmeyer, Judith Coenenberg, Anja Potthoff, Teferi Gebru Gebremeskel.

**Data curation:** Zekarias Gessesse Arefaine.

**Formal analysis:** Zekarias Gessesse Arefaine, Sintayehu Abebe, Ephrem Bekele, Amir Adem, Yohannes Adama, Norbert H. Brockmeyer.

**Investigation:** Zekarias Gessesse Arefaine.

**Methodology:** Zekarias Gessesse Arefaine, Yohannes Adama, Norbert H. Brockmeyer, Judith Coenenberg, Anja Potthoff.

**Software:** Zekarias Gessesse Arefaine, Yohannes Adama, Norbert H. Brockmeyer, Teferi Gebru Gebremeskel.

**Supervision:** Sintayehu Abebe, Ephrem Bekele, Amir Adem.

**Visualization:** Zekarias Gessesse Arefaine.

**Writing – original draft:** Zekarias Gessesse Arefaine, Sintayehu Abebe, Ephrem Bekele, Amir Adem, Yohannes Adama, Norbert H. Brockmeyer, Judith Coenenberg, Anja Potthoff, Teferi Gebru Gebremeskel.

**Writing – review & editing:** Zekarias Gessesse Arefaine, Sintayehu Abebe, Ephrem Bekele, Amir Adem, Judith Coenenberg, Anja Potthoff, Teferi Gebru Gebremeskel.

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
