## [Decision Letter · Decision Letter 0]

14 Feb 2020

Incidence and predictors of HIV related opportunistic infections after initiation of highly active antiretroviral therapy at Ayder Referral Hospital, Mekelle, Ethiopia: a retrospective single centered cohort study

PONE-D-19-24196

Dear Dr. Zekarias

We are pleased to inform you that your manuscript has been judged scientifically suitable for publication and will be formally accepted for publication once it complies with all outstanding technical requirements.

With kind regards,

Yatin N. Dholakia, MD

Academic Editor

PLOS ONE

Additional Editor Comments:

New guidelines for managing HIV infection recommend initiating ART as soon as one is diagnosed to have been infected.

The findings presented in the study may therefore have no relevance in the recent times, however they are interesting and may be used as baseline to any future similar study from the institution. The fact that OIs were more common in the first few months of initiating ART may be missing diagnosis before initiating ART or related to IRIS as the cases had low CD4 counts at initiation. This has been appropriately mentioned.

Journal Requirements:

1. Thank you for stating the following financial disclosure:

'The funders had no role in study design, data collection and analysis, decision to publish, or preparation of the manuscript.'

Please provide an amended Funding Statement that declares *all* the funding or sources of support received during this specific study (whether external or internal to your organization) as detailed online in our guide for authors at http://journals.plos.org/plosone/s/submit-now

Please state what role the funders took in the study.  If any authors received a salary from any of your funders, please state which authors and which funder. If the funders had no role, please state: "The funders had no role in study design, data collection and analysis, decision to publish, or preparation of the manuscript."

c. Please send your amended statements by return email; we will change the online submission form on your behalf.

2. Your ethics statement must appear in the Methods section of your manuscript. If your ethics statement is written in any section besides the Methods, please move it to the Methods section and delete it from any other section. Please also ensure that your ethics statement is included in your manuscript, as the ethics section of your online submission will not be published alongside your manuscript.

Reviewers' comments:

Reviewer's Responses to Questions

**Comments to the Author**

1. Is the manuscript technically sound, and do the data support the conclusions?

Reviewer #1: Yes

2. Has the statistical analysis been performed appropriately and rigorously? 

Reviewer #1: I Don't Know

3. Have the authors made all data underlying the findings in their manuscript fully available?

Reviewer #1: Yes

4. Is the manuscript presented in an intelligible fashion and written in standard English?

Reviewer #1: Yes

5. Review Comments to the Author

Reviewer #1: I find that this manuscript fulls all the criteria for publication. The only thing I would like to point out to the authors that it has been delayed for a long time. The analysis Tok place about 7 & 1/2 years back.

6. PLOS authors have the option to publish the peer review history of their article (what does this mean?). If published, this will include your full peer review and any attached files.

Reviewer #1: No

---

## [Editor Report · Acceptance letter]

3 Mar 2020

PONE-D-19-24196 

Incidence and predictors of HIV related opportunistic infections after initiation of highly active antiretroviral therapy at Ayder Referral Hospital, Mekelle, Ethiopia: a retrospective single centered cohort study 

Dear Dr. arefaine:

I am pleased to inform you that your manuscript has been deemed suitable for publication in PLOS ONE. Congratulations! Your manuscript is now with our production department. 

With kind regards,

on behalf of

Dr. Yatin N. Dholakia 

Academic Editor

PLOS ONE